# *N*-Glycosylation of LRP6 by B3GnT2 Promotes Wnt/β-Catenin Signalling

**DOI:** 10.3390/cells12060863

**Published:** 2023-03-10

**Authors:** Ruiyao Xu, Xianxian Wang, Sadia Safi, Nico Braunegger, Agnes Hipgrave Ederveen, Michelle Rottmann, Joachim Wittbrodt, Manfred Wuhrer, Janine Wesslowski, Gary Davidson

**Affiliations:** 1Institute of Biological and Chemical Systems-Functional Molecular Systems (IBCS-FMS), Karlsruhe Institute of Technology (KIT), 76344 Eggenstein-Leopoldshafen, Germany; 2Department of Immunology, University of Texas Southwestern Medical Center, Dallas, TX 75390, USA; 3Institute of Pharmacy and Molecular Biotechnology, University of Heidelberg, Im Neuenheimer Feld 364, 69120 Heidelberg, Germany; 4Center for Proteomics and Metabolomics, Leiden University Medical Center, Albinusdreef 2, 2333 ZA Leiden, The Netherlands; 5COS—Centre for Organismal Studies, Department of Molecular Developmental Biology & Physiology, Heidelberg University, Im Neuenheimer Feld 230, 69120 Heidelberg, Germany

**Keywords:** LRP6, Wnt signalling, Wnt/β-catenin, glycosylation, polylactosamine, B3GnT2

## Abstract

Reception of Wnt signals by cells is predominantly mediated by Frizzled receptors in conjunction with a co-receptor, the latter being LRP6 or LRP5 for the Wnt/β-catenin signalling pathway. It is important that cells maintain precise control of receptor activation events in order to properly regulate Wnt/β-catenin signalling as aberrant signalling can result in disease in humans. Phosphorylation of the intracellular domain (ICD) of LRP6 is well known to regulate Wntβ-catenin signalling; however, less is known for regulatory post-translational modification events within the extracellular domain (ECD). Using a cell culture-based expression screen for functional regulators of LRP6, we identified a glycosyltransferase, B3GnT2-like, from a teleost fish (medaka) cDNA library, that modifies LRP6 and regulates Wnt/β-catenin signalling. We provide both gain-of-function and loss-of-function evidence that the single human homolog, B3GnT2, promotes extension of polylactosamine chains at multiple *N*-glycans on LRP6, thereby enhancing trafficking of LRP6 to the plasma membrane and promoting Wnt/β-catenin signalling. Our findings further highlight the importance of LRP6 as a regulatory hub in Wnt signalling and provide one of the few examples of how a specific glycosyltransferase appears to selectively target a signalling pathway component to alter cellular signalling events.

## 1. Introduction

The Wnt pathway is an evolutionarily conserved, highly regulated cellular signalling network comprising 19 secreted Wnt proteins (ligands), 10 Frizzled proteins (principal receptors) and a variety of co-receptors that regulate embryonic development and tissue homeostasis [1,2,3]. Wnt signalling is traditionally classified as either β-catenin dependent (canonical) or β-catenin independent (non-canonical), and both established and emerging variations of Wnt signalling exist within each of these two major classes, in line with the enormous variety of associated biological responses [1,4,5,6,7]. Frizzled proteins are considered essential for transduction of almost all known Wnt signalling events and, similarly, the Wnt co-receptors LRP5/6 are considered essential for all β-catenin dependent Wnt signalling events. The key role played by LRP5/6 in Wnt/β-catenin signalling is well documented [8,9,10].

LRP5/6 are type I single-span transmembrane proteins with a large extracellular domain (ECD) harbouring four YWTD β-propeller (BP) domains, each linked to an epidermal growth factor-like (EGF-like) domain and three LDLA (low-density lipoprotein receptor type A) domains [8]. The four repeating units of the YWTD β-propeller-EGF-like (PE) domains form two rigid structural blocks, PE12 and PE34, which are connected by a flexible linker (hinge) that affords considerable bending flexibility to the ECD [11,12]. Wnt proteins as well as secreted Wnt inhibitors bind to the PE domains of LRP6 [12,13,14,15]. The intracellular domain (ICD) of LRP6 is rich in proline, serine and threonine residues and harbours five PPSPxS motifs that are phosphorylated by GSK3 and CK1 [16,17]. Phosphorylated PPSPxS motifs are reported to act as docking sites for Axin but also function as inhibitors of GSK3 kinase activity, thus promoting downstream Wnt/LRP signalling [18,19]. Like other membrane-spanning, cell surface proteins, LRP6 enters the secretory pathway in the endoplasmic reticulum (ER) and undergoes maturation processes as it passes through the ER and Golgi before localisation at the cell membrane. For LRP6, these maturation processes specifically require the chaperone MESD [20,21], which facilitates folding of the PE domains [22] as well as trafficking from the ER to Golgi and ultimately the plasma membrane [23]. In addition to phosphorylation, LRP6 undergoes palmitoylation [24] and is glycosylated at multiple sites within its ECD [11,12,25,26].

Addition of *N*-glycans to proteins in the ER starts with transfer of an immature, mannose-rich, dolichol-linked oligosaccharide. ALG8 is one of the enzymes involved in the biosynthesis of the dolichol-linked oligosaccharide precursor and has it been implicated in the regulation of Wnt/β-catenin signalling and LRP6 glycosylation [27]. Like many secreted and transmembrane proteins, LRP6 is detected as two bands in Western blot analysis due to its *N*-glycosylation state; the higher-molecular-weight band corresponds to mature LRP6 at the cell surface, whereas the lower-molecular-weight band corresponds to immature LRP6 within the ER/Golgi [20]. Although little is known with respect to the functional relevance of LRP6 glycosylation by specific glycosyltransferase enzymes, it has been reported that Mest/Peg1 prevents the maturation of LRP6 by controlling LRP6 glycosylation [25] and that fucosylation of complex *N*-glycans on LRP6 can regulate its endocytosis [28]. It has also been proposed that LRP6 lysosomal degradation is regulated by *O*-GlcNAcylation, acting as a nutrient sensing mechanism to reduce Wnt and increase Hippo signalling [29]. In addition, *N*-glycans close to the hinge have been reported to influence the bending angle of LRP6 ECD to regulate Wnt signalling [11].

*N*-glycans play important roles for their respective proteins, including trafficking, cell adhesion and signal transduction [30]. The β1,3-*N*-acetylglucosaminyltransferase (B3GnT) family includes type II glycosyltransferases that transfer *N*-acetylglucosamine (GlcNAc) via a β-1,3 linkage [31]. Prominent among the *N*-glycan modifications mediated by the B3GnT family is the synthesis of Poly-*N*-acetyllactosamine (polyLacNAc) chains. PolyLacNAc is composed of alternating residues of galactose (Gal) and *N*-acetylglucosamine (GlcNAc), linked β1→4 and β1→3, respectively. Initiation and elongation of polyLacNAc chain synthesis requires the activity of both B3GnT and β-1,4-galactosyltransferase (B4GALT) family members [32]. B3GnT2 catalyses both the initiation and elongation of polylactosamine chains and appears to recognise only the terminal disaccharide unit, thus it is capable of extending a variety of polylactosamine chain lengths [33,34]. B3GnT2 has been reported to be involved in the development of bladder transitional cell carcinomas, colon cancer and other gastrointestinal malignancies [35,36]. B3GnT2 null mice have significantly reduced levels of polylactosamine on *N*-glycans and show hyperresponsiveness of T cells, B cells and macrophages [37]. Single nucleotide polymorphisms (SNPs) in the B3GNT2 gene have been shown to reduce its expression and are associated with autoimmune diseases [38,39,40]. Moreover, a recent study identified several potential cancer cell receptor targets for B3GnT2 and additionally showed reduced interaction between some ligands and receptors between tumour and T-cells upon overexpression of B3GnT2 [41].

Here, we identify B3GnT2 as a post-translational modifier and positive regulator of LRP6, which directly links a specific glycosyltransferase acting on protein *N*-glycans to a receptor that transduces Wnt signals.

## 2. Materials and Methods

### 2.1. Antibodies and Plasmids

Rabbit polyclonal T1479 antibody was as described [17]. Other antibodies used were as follows: mouse monoclonal anti-LRP6; rabbit polyclonal anti-Wnt3a (Abcam, Cambridge, UK); rabbit polyclonal anti-vinculin (Cell Signaling Technology, Leiden, the Netherlands); mouse monoclonal anti-V5 and anti-Flag M2 affinity gel (Sigma-Aldrich, Taufkirchen, Germany).

LRP6 and the LRP6 N→Q mutants N433Q and N486Q were cloned from *Flag-PAR4-hLRP6*, *Flag-PAR4-hLRP6-(△ig1)N433Q* and *Flag-PAR4-hLRP6-(△og1)N486Q*, respectively (kindly provided by Junichi Takagi) [11]. Other *hLRP6* N-Q mutants were generated with the QuikChange Lightning Site-Directed Mutagenesis Kit (Agilent Technologies, Waldbronn, Germany). The *pCMV-Sport6.1-OlB3GnT2-like* plasmid was isolated from the medaka cDNA library and sequence verified (gene ID: 101167058). Human *B3GnT2* was cloned from *hB3GnT1* (*hB3GnT2*) human ORF clone (NM_006577, OriGene Technologies, Herfurt, Germany) and human *B3GnT8* was cloned from *hB3GnT8* ORF clone (NM_198540, OriGene Technologies, Herfurt, Germany) into *pCS2^+^* expression vectors. Human *B3GnT1*, *B3GnT3*, *B3GnT4*, *B3GnT5*, *B3GnT6*, *B3GnT7* and *B3GnT9* coding sequences were ordered as codon-optimised linear dsDNA fragments (Thermo Fisher Scientific, GeneArt, Darmstadt, Germany) and cloned into *pCS2^+^* expression vectors. *HiBiT-hLRP6-mCherry* was cloned from human *pCS2^+^ hLRP6-mCherry* previously described in [42]. To fuse the coding sequence of *HiBiT* N-terminally to *hLRP6-mCherry*, a complementary pair of ssDNA was ordered (Metabion, Planegg/Steinkirchen, Germany) consisting of an LRP6 signal peptide sequence followed by the HiBiT sequence. The two ssDNAs were annealed and fused at the N-terminus to *hLRP6-mCherry.* All constructs were validated by sequencing (Microsynth Seqlab, Göttingen, Germany). *pCDNA3 hβ-catenin* was a gift from Eric Fearon (Addgene # 16828, Watertown, MA, USA). Other plasmids were previously described [17].

### 2.2. Medaka (Oryzias latipes) cDNA Library Screening for LRP6 Modifiers

A medaka cDNA library was used to screen for regulatory modifiers of LRP6 [43]. A total of 24 cDNA clones were mixed as a pool and 740 pools of potential LRP6 modifiers were prepared as transfection-ready plasmid DNA samples. Medaka cDNA pools (120 ng) were co-transfected with *pCS2^+^ 6myc-hLRP6* (10 ng) into HEK293T cells in 96-well plates. Then, 20 h after transfection, cells were lysed in 1% triton lysis buffer (1% Triton X-100, 50 mM Tris-HCl (pH7.4), 150 mM NaCl, 5 mM Na3VO4, 25 mM NaF, 0.1% NP-40, 1 mM EDTA (pH 8.0), 1 mM EGTA, supplemented with protease inhibitor cocktail (Roche), adjusted pH to 7.0) and subjected to SDS-PAGE/Western blot to detect LRP6 modification. Secondary screening of individual clones within candidate pools was performed to identify the specific clone conferring LRP6 modification.

### 2.3. Cell Culture, Cell Transfection

All cells were maintained at 37 °C and 5% CO_2_ in Dulbecco’s modified Eagle medium (DMEM, Fisher Scientific, Schwerte, Germany) containing 10% fetal bovine serum (FBS, Fisher Scientific, Schwerte, Germany) and 1% penicillin-streptomycin (P/S; Gibco; Fisher Scientific, Schwerte, Germany). Human embryonic kidney 293T (HEK293T) cells (ATCC^®^ CRL-3216, Manassas, VA, USA) were transfected with plasmid DNA using either PromoFectin (PromoCell) or ScreenFect^®^A (ScreenFect, Eggenstein-Leopoldshafen, Germany) according to the manufacturer’s protocols. The reverse transfection method (one-step transfection, combined plating and transfection) was used.

### 2.4. TOPFLASH Reporter Assay

Cells were transfected in 96-well plates in tetraplicates with TOPFLASH/Renilla plasmids (20/2 ng) and harvested 48 h post-transfection in passive lysis buffer (Promega, Walldorf, Germany). All error bars shown are standard deviation (SD) from mean of ≥ 3 replicates. Two-tailed Student’s t-test was performed to compare the difference of the two groups (* *p* < 0.05; ** *p* < 0.01; *** *p* < 0.001; **** *p* < 0.0001).

### 2.5. Western Blot, Immunoprecipitation (IP), Lectin Blot

Cells were lysed in 1% triton lysis buffer. For untagged LRP6 protein, IP was performed with mouse monoclonal anti-LRP6 (Abcam, Cambridge, UK) and protein G agarose (Thermo Fisher Scientific, Idstein, Germany). For flag-tagged LRP6 protein, IP was performed using anti-Flag M2 affinity gel (Sigma-Aldrich, Taufkirchen, Germany). IP products were used for lectin blotting. The PVDF membrane (Bio-Rad Laboratories, Feldkirchen, Germany) was blocked with RIPA buffer (50 mM Tris-HCL, 0.1% Trito X-100, 0.1% Sodium Deoxycholate, 150 mM NaCl, 0.1% SDS, adjusted to pH 7.4) for 1 h at room temperature. Amounts of 2 μg/mL biotinylated Lycopersicon esculentum lectin (LEL, Vector Laboratories) or 5 μg/mL biotinylated Concanavalin A (ConA) (Vector Laboratories, Newark, CA, USA) were prepared freshly in RIPA buffer and incubated with the membrane at 4 °C overnight. After washing with RIPA buffer, the membrane was incubated with 1/2500 streptavidin-HRP (GE HealthCare, Braunschweig, Germany) for 2 h at room temperature. The membrane was washed with TBST (50 mM Tris, 150 mM NaCl, 0.05% Tween-20, adjusted to pH 8.0) buffer before subjecting to ECL detection.

### 2.6. RNA Extraction, RT-PCR, Semi-qPCR

Cells were lysed with TRIzol reagent (Invitrogen, Fisher Scientific, Schwerte, Germany) and RNA isolation was performed according to the manufacturer’s protocol. DNase I (Promega, Walldorf, Germany) was used to digest contaminated DNA. Then, 1st strand cDNA was synthesised with a random primer (Promega, Walldorf, Germany) and MLRVT kit (Promega, Walldorf, Germany). Using 1st strand cDNA as the template, semi-qPCR was performed with Go Taq DNA polymerase (Promega, Walldorf, Germany) and designed primers, β-actin (Forward: AGC ATC CCC CAA AGT TCA CAA; Reverse: GCT ATC ACC TCC CCT GTG TGG) and B3GnT2 (Forward: AGG CAT ACT GGA ACC GAG AG; Reverse: GAT GGC TTA TAT TGG AGA GCC TG).

### 2.7. Immunofluorescence Analysis

For immunofluorescence analysis, cells were seeded in 18-well μ-slides (ibidi, Gräfelfing, Germany). The following day, cells were fixed in 4% paraformaldehyde/Dulbecco’s PBS +/+ (Gibco, Fisher Scientific, Schwerte, Germany) for 10 min at room temperature and then treated with 20 µg/mL FITC-LEL (Vector Laboratories Inc., Newark, CA, USA) at room temperature for 1 h. After washing with Dulbecco’s PBS+/+ (Gibco, Thermo Fisher), cells were imaged using a ZEISS LSM 800 confocal fluorescence microscope (Zeiss, Jena, Germany) fitted with an LD LCI Plan-Apochromat 40x/1.2 oil differential interference contrast (UV) VIS-IR objective (Zeiss, Jena, Germany). The FICT-labelled LEL was excited at 488 nm and the emission was captured in the range of 482–631 nm using the GaAsP-PTM detector. Images were analysed using Fiji [44].

### 2.8. Mass Spectrometry Analysis

HEK293T cells were transfected with 2.5 μg *pCS2^+^ hLRP6* +/−1.5 μg *pCMV-Sport6.1 OlB3GnT2l*. Small aliquots of lysates and immune-purified (IP) LRP6 were analysed by SDS-PAGE/Western blot and the remaining IP was separated by SDS-PAGE and stained with Coomassie (50% (*v/v*) methanol, 10% (*v/v*) acetic acid, 0.5% CBB R250). The excised gel bands corresponding to different forms of the LRP6 protein were subjected to in-gel trypsin digestion. The tryptic digests from the band excisions were separated and analysed by a C18 reversed-phased (RP)-LC-MS/MS. The system was equipped with an Acclaim PepMap 100 trap column (100 μm × 20 mm, particle size 5 μm, Thermo Scientific, Waltham, MA, USA) and an Acclaim PepMap RSLC C18 nano-column (75 μm × 150 mm, particle size 2 μm, Thermo Scientific). One microliter of sample (20 ng) was injected and the (glyco)peptides were separated with a gradient from 99% solvent A (0.1% formic acid in water) and 1% solvent B (95% ACN) to 50% solvent B over 30 min, with a flow rate of 700 nL/min. The nanoLC was coupled to a maXis quadrupole time-of-flight-MS (q-TOF-MS; Bruker Daltonics, Bremen, Germany) equipped with a nanoBooster (Bruker Daltonics, Bremen, Germany). Tandem mass spectra data were acquired at 1 Hz from *m/z* 50–2800 with the precursor selection above *m/z* values of 500, excluding singly charged ions, and the stepping mode was applied for the tandem MS collision energy (10.1074/mcp.RA117.000240).

### 2.9. Cell Surface LRP6 Nano-BiT Assay

A total of 3.96 × 10^6^ HEK293T cells were transfected with 20 ng *pCS2^+^ HiBit-hLRP6-mCherry*, 5 ng *pCMV-Sport6.1* m*Mesd* and 0.5 ng of *pCS2^+^ hB3Gnt2* using ScreenFect^®^A. Cells were seeded in poly-d-lysine (Gibco, Fisher Scientific, Schwerte, Germany)-coated white-walled 96-well cell culture plates with clear bottom (VWR part of avantor, Darmstadt, Germany). Forty-eight hours post-transfection, cells were washed once with 200 µL of Hanks’ balanced salt solution (HBSS; Sigma-Aldrich, Taufkirchen, Germany) and incubated with 90 μL of detection reagent (LgBiT 1:200 (Promega, Walldorf, Germany), Furimazine 1:100 (Promega, Walldorf, Germany), HEPES 1:100 (Gibco, Fisher Scientific, Schwerte, Germany) and 5% FBS in phenol red free DMEM medium) for 10 min at 37 °C without CO_2_. The bioluminescence to fluorescence ratio was measured using a CLARIOstar Plus microplate reader (BMG LABTECH, Ortenberg, Germany).

### 2.10. Lentiviral-Mediated B3GnT2 Gene Knockout

#### 2.10.1. sgRNA Design for Inactivation of B3GnT2 in HEK293T Cells

A pair of sgRNAs flanking the B3GnT2 CDS region were selected using the online CRISPOR tool (http://crispor.tefor.net) and synthesised by Metabion (Planegg/Steinkirchen, Germany). The sgRNA sequences are:

*B3GnT2*-sgRNA-For: CACC GCTG TAAA CCAC TATT CCTG

*B3GnT2*-sgRNA-Rev: CACC GTTC TGAT CTTA CCGG CTAG

#### 2.10.2. Cloning Lenti-CRISPRV2-B3GnT2-sgRNA Plasmid

The *B3GnT2* sgRNA oligo pairs were phosphorylated using T4 PNK (Fisher Scientific, Schwerte, Germany) and annealed. The annealed upstream *B3GnT2*-sgRNA-For sgRNAs were ligated into BsmBI-digested l*entiCRISPRv2* and the downstream *B3GnT2*-sgRNA-Rev sgRNAs were ligated into *lentiCRISPRv2-blast* vectors using T4 ligase (Fisher Scientific, Schwerte, Germany). *LentiCRISPRv2* was a gift from Feng Zhang (Addgene #52961, Watertown, MA, USA). *LentiCRISPRv2-blast* was a gift from Brett Stringer (Addgene, # 98293, Watertown, MA, USA).

#### 2.10.3. Infecting HEK 293T Target Cells with LentiCRISPR-B3GnT2-sgRNA Lentivirus

To package *LentiCRISPRV2-B3GnT2-sgRNA* into lentiviruses, HEK 293T cells at 80% confluence were prepared in 10 cm dishes. At 1 h before transfection, 5 mL of culture medium was replaced by fresh DMEM supplemented with 10% FBS and 1% P/S. Two plates of cells were then transfected with envelope plasmid *pCMV-VSVG* (3 μg), packaging plasmid *psPAX2* (4,5 μg) and either *LentiCRISPRV2-B3GnT2-sgRNA-For* (7.5 µg) or *LentiCRISPRV2-B3GnT2-sgRNA-Rev* (7.5 µg) using ScreenFect^®^A (Screenfect, Eggenstein-Leopoldshafen, Germany) according to the manufacturer’s protocol and incubated at 37 °C with 5% CO_2_. At 24 h post-transfection, the cell culture medium of both plates was collected and passed through a 0.22 μm filter before adding both filtrates simultaneously to HEK 293T cells at 50% confluency. An amount of 5 mL fresh DMEM supplemented with 10% FBS and 1% P/S was then added to the lentiviral packing cells and incubated for another 24 h. The cell culture medium was collected again and used to infect the HEK 293T target cells for a second time with both filtrates. Infected cells were incubated for a further 24 h before initiating a 2-week antibiotic selection procedure using Puromycin (2 µg/mL) and Blasticidin (5 µg/mL). *pCMV-VSV-G* was a gift from Bob Weinberg (Addgene #8454, Watertown, MA, USA). *psPAX2* was a gift from Didier Trono (Addgene #12260, Watertown, MA, USA).

#### 2.10.4. Generation of Single-Cell Clones and Genotyping

After antibiotic selection, surviving cells were dispersed with trypsin and diluted to 0.3 cells/100 µL, seeded in 96-well plates and incubated at 37 °C with 5% CO_2_ for 7 days. Single-cell colonies were carefully selected using bright field microscopy and gradually scaled up to 2 × 6 cm cell culture dishes. One dish was kept for passaging and the other was used for genomic DNA extraction using the E.Z.N.A.^®^ Tissue DNA KiT (Omega Bio-tek, Norcross, GA, USA) to perform PCR-based genotyping. This was performed using Q5 high-fidelity DNA polymerase (NEB, Frankfurt am Main, Germany) with the following PCR primers:

*B3GnT2-ATG-sc-For*: CCT TTG AAG TGG GCT TGA

*B3GnT2-EXON-sc-Rev*: GCC AAT AGG TTT ATA AAT ACC TGA ATT

#### 2.10.5. Sanger Sequencing of B3GnT2 Gene KO Cell Clones

To validate the correctness of the *B3GnT2* gene KO clone, a PCR product covering the edited genomic region was amplified and purified using the peq GOLD MicroSpin cycle Pure kit (PEQLAB, VWR part of avantor, Darmstadt, Germany). The purified PCR product was sent for Sanger sequencing (Microsynth Seqlab, Göttingen, Germany) premixed with the PCR primer *B3GnT2-ATG-sc-For*.

## 3. Results

### 3.1. Identification of Medaka B3GnT2l as A Novel LRP6 Modifier

Due to the evolutionarily conserved nature of Wnt signalling, a variety of different cell lines and tools from different species may be used to study the pathway, an approach we have taken advantage of previously [17,45,46,47]. Here, an arrayed and annotated medaka (Oryzias latipes) cDNA library [43] was used as a source of genes to screen for functional regulators of human LRP6 in a human cell line (HEK293T) (Figure 1a). Cells were co-transfected with expression plasmids encoding human LRP6 and pools of 24 cDNA clones prepared from the annotated medaka library and arrayed in 96-well plates ready for transfection. Whole cell lysates were prepared 24 h post-transfection and the overexpressed LRP6 proteins were analysed by SDS-PAGE/Western blot. One of the cDNA pools (96-well plate Nr. 7, well position H9) conferred a clear upshift of the upper LRP6 band (Figure 1a,b). In a second round of screening, the 24 cDNA clones present in pool 7H9 were analysed individually and the medaka B3GnT2-like gene (Ol B3GnT2l, gene ID: 101167058) was identified as the sole modifier (Figure 1c). B3GnT2l is a member of the B3GnT family of glycosyltransferases that plays roles in the N-glycosylation of proteins and, in particular, in the biosynthesis of poly-N-acetyllactosamine chains [48,49,50,51] (Appendix A).

TOPFLASH Wnt reporter assays demonstrated that the upshift of the LRP6 protein upper band seen upon co-expression of B3GnT2l with LRP6 coincides with enhancement of Wnt/β-catenin signalling (Figure 1d, lanes 2,3). No effect was seen when B3GnT2l was co-expressed with a truncated form of LRP6 lacking most of its extracellular domain (LRP6ΔE1-4) and that functions as a constitutively active form of the receptor [10,52] (Figure 1d, lanes 4,5). This confirms that functional modification is specific to the ECD region of LRP6, which is known to harbour all N-glycosylation sites. In agreement, LRP6ΔC, a dominant negative form of LRP6 that only harbours the ECD and TM domains [10], was modified by B3GnT2l (Figure 1d, lanes 6,7). These results show that B3GnT2l promotes LRP6 modification, thereby activating Wnt/β-catenin signalling. No other plasmid cDNA from pool H9 in plate #7, when co-expressed with LRP6, had any significant effect on the Wnt co-receptor protein or Wnt/β-catenin signalling.

To confirm that B3GnT2l acts directly on *N*-glycans attached to LRP6, cell lysates were treated with peptide:N-glycosidase F (PNGase F), which removes all *N*-glycans attached to glycoproteins [53]. As expected, PNGase F treatment resulted in the formation of a single, lower-molecular-weight LRP6 core protein band as seen by SDS-PAGE/WB and no remnants of an upshifted band could be seen (Figure 1e, PNGaseF). Similar results were seen if cells were first treated with the global inhibitor of *N*-glycosylation, tunicamycin (Appendix A), confirming that medaka B3GnT2l modifies the *N*-glycans of LRP6. The upper and lower bands of LRP6 seen in immunoblots represent the mature cell surface form of the receptor and the immature ER form, respectively. The fact that B3GnT2l appears to modify predominantly the upper band of LRP6 suggests it acts on more mature *N*-glycan chains attached to the Wnt co-receptor, which are more complex in nature. In agreement with this, treatment of cell lysates with endoglycosidase H (EndoH), which specifically cleaves mannose-rich immature *N*-glycans [54], targeted immature LRP6 (lower bands) but failed to remove either the upper band of LRP6 or the upshifted upper band from cells overexpressing LRP6 and B3GnT2l (Figure 1e, EndoH). Experiments using the exoglycosidase neuraminidase, which specifically cleaves terminal sialic acid residue on complex N-glycans, also support this (Appendix A). In agreement with these results, co-IP experiments demonstrated interaction between B3GnT2l and LRP6 (Appendix A). In line with B3GnT2l activating Wnt/β-catenin signalling at the level of Wnt reception at the membrane, its enhancing effect was far more significant when the pathway was activated upstream of β-catenin (Figure 1f). Taken together, the above results suggest that B3GnT2l, when co-expressed with LRP6 in cells, affects glycosylation of complex *N*-glycans on the extracellular domain of LRP6, thereby promoting its ability to transduce Wnt/β-catenin signalling.

### 3.2. Human B3GnT2 Promotes Wnt/β-Catenin Signalling and Acts at the Level of LRP6

Although Oryzias latipes (Ol) B3GnT2l was identified in the modification screen, characterisation of the human homolog(s) is more relevant for studying the cellular and molecular mechanisms underlying this functional modification of LRP6 in human cells. Among the nine human B3GnT family members [48,49,50,51], B3GnT2 and B3GnT8 have the highest sequence identity to Ol B3GnT2l, sharing 34.2% and 33.5%, respectively (Appendix A). Western blot and TOPFLASH Wnt reporter assays were performed to confirm whether these human homologs of Ol B3GnT2l confer similar functional modification on LRP6. Human B3GnT2 enhanced Wnt/β-catenin signalling and modified (upshifted) the LRP6 upper band (Figure 2a), whereas human B3GnT8 had no obvious effect on either (Figure 2b). This demonstrates not only an evolutionarily conserved property for a B3GnT family member but also specificity within the B3GnT family for regulating LRP6 function. It is reported that B3GnT2 and B3GnT8 act cooperatively to promote protein glycosylation [55]; however, no synergy between these glycosyltransferases was seen with respect to modification of LRP6 or activation of Wnt/β-catenin signalling (Appendix A). Indeed, none of the other B3GnT family members, when co-expressed with B3GnT2, led to a significantly stronger activation of Wnt/β-catenin signalling (Appendix A).

Similar to the results we obtained for Ol B3GnT2l, h B3GnT2 had no significant effect when Wnt/β-catenin signalling was activated downstream of LRP6, at the level of β-catenin (Figure 2c), in line with it acting at the level of LRP6 to regulate Wnt/β-catenin signalling. In addition to the Wnt co-receptor LRP6, secreted Wnt proteins as well as their main receptors, the Frizzled (FZD) proteins, are *N*-glycosylated. We therefore tested if B3GnT2 acts on either of these Wnt pathway components to promote Wnt/β-catenin signalling; however, only LRP6, and not FZD5, FZD8 or Wnt3a, was modified upon co-expression of human B3GnT2 (Figure 2d). Immunoblots using an antibody that specifically recognises phosphorylated PPSP motifs within the intracellular domain of LRP6 [16,17] also demonstrated that the TOPFLASH signalling induced by B3GnT2l is not mediated by LRP6 phosphorylation (Appendix A). These results suggest that, within the Wnt/β-catenin pathway, human B3GnT2 acts specifically on LRP6 *N*-glycans to promote signalling at the cell membrane.

### 3.3. B3GnT2 Extends Polylactosamine Chains of LRP6 N-glycans

B3GnT2 is one of the major glycosyltransferases involved in polylactosamine synthesis [37]. Polylactosamine consists of repeating N-acetyllactosamine disaccharide units (Galβ1–4GlcNAcβ1–3)_n_ and is commonly referred to as polyLacNAc. It is synthesised by the coordinate action of a β1,4-galactosyltransferase and a β1,3-N-acetylglucosaminyltransferase. To investigate whether B3GnT2 modifies LRP6 by promoting the synthesis of polylactosamine chains on its *N*-glycans, we performed lectin blots on immunoprecipitates (IPs) from cells expressing LRP6 alone or co-expressed with either Ol B3GnT2l, h B3GnT2 or h B3GnT8. We employed the Lycopersicum esculentum lectin, LEL/LEA (commonly known as tomato lectin) as a specific probe for polyLacNAc extensions [56] and the broadly selective mannose lectin Concanavalin A (ConA) as a control for detecting *N*-glycosylated LRP6. In contrast to overexpression of LRP6 alone, which showed only a weak signal for polyLacNAc, co-expression of either Ol B3GnT2l or h B3GnT2 promoted clear polylactosamine synthesis on LRP6 (Figure 3a, LEL lectin blot). The LEL blot signal was stronger for Ol B3GnT2l, which is in line with our general observation that Ol B3GnT2l promotes a stronger and higher molecular weight upshift of the LRP6 upper band compared to h B3GnT2 (Figure 3a, lower blot and Appendix A). These results confirm that the modification of LRP6 mediated by B3GnT2 is most likely due to the synthesis of polylactosamine chains. In contrast to B3GnT2, B3GnT8 failed to promote polyLacNAc synthesis on LRP6 beyond the low background levels (Figure 3a). We also tested whether other family members could synthesise polyLacNAc on LRP6 *N*-glycans and, in addition to B3GnT2, found that somewhat lower LEL signals were also detected for B3GnT3, B3GnT4 and B3GnT9 (Appendix A). However, TOPFLASH Wnt reporter assays and LRP6 Western blot analysis of all nine members demonstrated that only B3GnT2 had the ability to simultaneously modify LRP6 *N*-glycans and activate Wnt/β-catenin signalling (Appendix A). Although B3GnT3, 4 and 9 showed some polyLacNAc synthesis activity towards LRP6, albeit lower than B3GnT2, they either had no significant effect (B3GnT4) or an inhibitory effect (B3GnT3 and 9) on LRP/Wnt signalling (Appendix A). Interestingly, B3GnT7, which is reported to synthesise keratin sulphate (KS) by adding N-acetylglucosamine to KS glycosaminoglycans [57,58,59], showed the strongest inhibition of LRP6/Wnt signalling of any family member and induced a weakly diffuse but large upshift of the upper LRP6 protein band together with a strong LEL signal (Appendix A). Nevertheless, B3GnT7 failed to significantly alter B3GnT2-induced LRP6/Wnt signalling when compared to other members (Appendix A).

To confirm the structural modification of LRP6, reversed-phased liquid chromatography-mass spectrometry (LC-MS) glycopeptide analysis was performed. Analysis was first performed using immunoprecipitated (IP) Flag-LRP6, obtained from lysates of HEK293T cells overexpressing either LRP6 alone or LRP6 co-expressed with the Ol B3GnT2-like gene (Figure 3b). Modification of LRP6 *N*-glycans attached to positions N81 and N433 were observed, indicated by the signals observed at m/z 1483.876, 1551.571, 1605.586, 1673.280 and 1727.294 (Figure 3c, Region N81) as well as 1331.162, 1379.848 and 1501.557 (Figure 3c, Region N433). The MS result demonstrated that glycopeptide fragments derived from the *N*-glycans attached to Asn81 (N81) and Asn433 (N433) on the extracellular domain (ECD) of LRP6 had polylactosamine extensions (Figure 3c, red dashed box within mass spectrum). Tandem MS analysis showed that a diantennary glycan is attached to N433, and di-LacNAc and sialylated di-LacNAc fragments were observed at m/z 731.264 and 1022.354, respectively (Appendix A). We also attempted LC-MS analysis of LRP6 after co-expression with human B3GnT2; however, insufficient yields of the purified, upshifted LRP6 protein were obtained during sample preparation and we were unable to detect modification. These results nevertheless confirm that Ol B3GnT2l extends the polylactosamine chains of *N*-glycans attached to LRP6, at least at position N433.

### 3.4. Multiple N-glycan Sites of LRP6 Are Relevant for B3GnT2-Mediated LRP6 Modification

There are 10 putative N-X-S/T (X ≠ P) *N*-glycosylation sites located in the ECD of LRP6, 4 of which are evolutionarily conserved as they are also present in the Drosophila LRP6 homolog, Arrow (Figure 4a). These sites are distributed relatively evenly throughout the PE domains (Figure 4b). To identify which LRP6 *N*-glycan sites are relevant for B3GnT2-medated regulation of LRP6/Wnt signalling, we generated asparagine to glutamine (N to Q) mutants at all 10 sites. We first expressed each of the *N*-glycosylation mutants in HEK293 cells, with and without co-expression of B3GnT2, and the corresponding LRP6 proteins were analysed using SDS-PAGE/WB (Figure 4c). Most LRP6 mutants proteins were expressed at similar levels and upper:lower band ratios were also similar in most cases when compared to wild-type (WT) LRP6 (Figure 4c). However, for two of them (N81Q and N692Q), no upper band representing the mature form of the receptor could be detected (Figure 4c). This suggests that *N*-glycans attached to N81 and N692 of LRP6 are required for full maturation and cell surface trafficking of the Wnt co-receptor. It is noteworthy that both of these sites are evolutionarily conserved in LRP6 (Figure 4a). Not surprisingly, both LRP6^N81Q^ and LRP6^N692Q^ displayed very weak activation of Wnt/β-catenin signalling when analysed using TOPFLASH reporter assays (Figure 4d), confirming their crucial role in the biogenesis of fully functional LRP6. LC-MS glycopeptide analysis also demonstrated clear Ol B3GnT2-like-mediated extension of polylactosamine chains of the *N*-glycan attached to N81 (Figure 4c). In contrast, LRP6^N859Q^ displayed significantly enhanced levels of Wnt/β-catenin signalling when overexpressed (Figure 4d), indicating that the *N*-glycan attached to N859 functions to inhibit receptor signalling. Interestingly, although the WB results showed no obvious increase in the mature LRP6^N859Q^ band (Figure 4c), LEL lectin blots showed that it had significantly higher levels of polylactosamine (Appendix A). This fits with our hypothesis that extension of polylactosamine units of LRP6 *N*-glycans enhances the signalling activity of LRP6. The other seven LRP6 mutants transduced Wnt/β-catenin signalling at levels approximating that of wild-type LRP6 (Figure 4d), in line with their expression (Figure 4c). To address the functional consequences of specific *N*-glycan ablation on LRP6 for B3GnT2-mediated activation of Wnt/β-catenin signalling, TOPFLASH reporter assays were performed upon co-expression of the LRP6 mutants with B3GnT2 (Figure 4e). Excluding N81Q and N692Q, which fail to mature into functional receptors, no LRP6 mutant displayed a markedly altered ability to be activated by B3GnT2 when compared to LRP6 WT (Figure 4e). These results are in general agreement with our WB analysis, where no marked change in the ability of B3GnT2 to promote modification of the upper LRP6 band could be observed (Figure 4c). Taken together, these results demonstrate that B3GnT2 most likely does not have a preference for specific *N*-glycans on LRP6 but rather acts on multiple *N*-glycans to regulate its Wnt/β-catenin signalling activity.

### 3.5. B3GnT2 Promotes Cell Surface Levels of LRP6

We next asked whether B3GnT2-mediated modification of *N*-glycans on LRP6 functions to alter its cell surface trafficking, which is a known function of protein glycosylation and would provide an explanation for its ability to regulate LRP6/Wnt signalling. To study this, we used the HiBiT protein tagging system. HiBiT (High BiT) is an 11 amino acid peptide tag that can be attached to any protein-of-interest and is detected via a simple bioluminescent assay upon association with LgBiT (Large BiT) to reconstitute an active Nano-luciferase (Nluc) enzyme (Figure 5a) [60]. In order to specifically detect cell surface HiBiT-tagged LRP6, cell-impermeable LgBiT was added to cell cultures together with the substrate for Nluc. Experiments were performed with or without over-expression of Mesd, which is known to promote cell surface LRP6 levels and, like B3GnT2, acts in the Golgi. Similar to Mesd, B3GnT2 increased cell surface levels of LRP6 approximately twofold (Figure 5b). When both Mesd and B3GnT2 were co-expressed with LRP6, the cell surface levels were further enhanced (Figure 5b). These results indicate that B3GnT2 functions, at least in part, to promote trafficking of LRP6 from the Golgi to the cell surface.

### 3.6. Human B3GnT2 Is Required for Wnt/β-Catenin Signalling

Our gain-of-function experiments suggest that B3GnT2 promotes Wnt signalling via extension of polylactosamine chains on the *N*-glycans attached to LRP6, which enhances its cell surface localisation and therefore its ability to transduce Wnt/β-catenin signals. To address the requirement of B3GnT2 for Wnt/LRP6 signalling, we generated B3GnT2−/− HEK293T cell lines using a lentiviral-based CRISPR/Cas9 gene editing approach designed to remove the entire B3GnT2 coding region (Appendix A, see Materials and Methods for details). Since no suitable antibody could be identified for the detection of endogenous B3GnT2, we confirmed B3GnT2 gene disruption using both semi-qPCR (Appendix A) and genomic sequencing (Appendix A). Loss of B3GnT2 in HEK293 cells strongly reduces the global level of polylactosamine present at the cell surface (Figure 5c). This suggests that B3GnT2 is the predominant enzyme responsible for polylactosamine synthesis in these cells. If B3GnT2 plays a functional role in LRP/Wnt signalling, then its ablation would be expected to reduce LRP6-mediated signalling, and this is indeed the case. Specifically, TOPFLASH reporter assays showed a reduction of Wnt/β-catenin signalling in B3GnT2−/− cells compared to wild-type (WT) cells, which was true for both basal signalling (Figure 5d, lanes 1,2 upper graph) as well as LRP6-induced signalling (Figure 5d, lanes 3,4 upper graph, Appendix A). This reduction was more apparent upon LRP6 over-expression (Figure 5d, upper graph, compare lane 4 vs. 3 with lane 2 vs. 1). The reduction in Wnt/β-catenin signalling is not due to a reduction in the over-expressed levels of LRP6 protein; indeed, LRP6 levels are somewhat higher in B3GnT2−/− cells (Figure 5d, lower blots, lanes 3–4). Importantly, we could show that the observed reduction in Wnt signalling in B3GnT2−/− cells is specifically due to loss of B3GnT2, as the effect could be rescued upon re-expression of B3GnT2 (Figure 5d, upper graph, lanes 5–7). An upshift of the mature LRP6 protein band was also seen upon co-expressing LRP6 with increasing amounts of B3GnT2 in B3GnT2−/− cells (Figure 5d, lower graph, lanes 4–7). Taken together, the results presented here show that B3GnT2 functions to promote the synthesis of polylactosamine units on multiple *N*-glycans of LRP6 to activate Wnt/β-catenin signalling.

## 4. Discussion

Approximately 200 glycosyltransferases and additional glycan-modifying enzymes act on glycoproteins to diversify the cellular proteome [61]. The complex structural and functional properties of glycans on proteins, however, makes it challenging to assign functional specificity to individual glycosyltransferases. In this study, we have provided both gain-of-function and loss-of-function evidence that a single member of the human B3GnT family, namely B3GnT2, modifies multiple *N*-glycans of LRP6 via elongation of polylactosamine to positively regulate Wnt/β-catenin signalling. LRP6 undergoes *N*-glycosylation at several sites adjacent to the ligand binding regions within the β-propeller domains [12,23] as well as near the ECD flexible hinge region that are proposed to regulate LRP6 conformations [11]. The effect these N-glycan moieties may have on interaction partners has not been studied in great detail. Our analysis has identified trafficking of mature LRP6 to the cell surface as the most likely mechanism by which B3GnT2 helps promote Wnt/β-catenin signalling. Nevertheless, considering the number of *N*-glycans on the ECD of LRP6 that appear to be targeted by B3GnT2, it will be of interest to perform additional studies in the future to address whether the interactions of LRP6 with its multiple ligands, or other receptors, are affected. Likewise, our finding that additional B3GnT family members appear to modify LRP6 via polylactosamine synthesis, which can alter its Wnt signalling activity in different ways, indicates that this family of glycosyltransferases may play a more complex role in LRP6/Wnt signalling.

The *Oryzias latipes* (*Ol*) *B3GnT2-like* cDNA identified in this study may have arisen from a Teleost genome duplication event around 350 million years ago that generated extra copies of fish genes [62]. Surprisingly, closer inspection of the medaka cDNA library showed that only *OlB3GnT2-like* gene was present as a full-length cDNA clone. Considering human B3GnT2 appears to be the functional homolog but displays a weaker overall effect on LRP6 compared to the OlB3GnT2-like gene, this raises the question if either of the two *OlB3Gnt2* genes (*OlB3Gnt2a/b*) may have the same, or indeed even a stronger, effect on LRP6. It may be of interest to study this in the future to try and understand if (1) the medaka B3GnT2-like gene is the only functional homolog of hB3GnT2 and (2) why the enzymatic activity of the medaka glycosyltransferase appears to be higher than the human homolog.

As both B3GnT2 and B3GnT8 have been reported to synthesise polylactosamine chains and are the closest homologs of OlB3GnT2l, this raises the question of what determines the apparent specificity of B3GnT2 for LRP6. In addition to B3GnT2 and 8, other members have been implicated in the synthesis of polyLacNAc chains, such as B3GnT3 [63], and we have also shown that B3GnT3, 4, 7 and 9 can synthesise polyLacNAc on LRP6. It remains unclear why B3GnT7, which is reported to be specific for sulphated LacNAc chains present in keratin sulphate glycosaminoglycans [57,59], should stain positive for the polyLacNAc selective lectin, LEL. Like B3GnT2, which recognises and elongates LacNAc disaccharides in conjunction with B4GalT family members, B3GnT7 elongates LacNAc disaccharides; however, the N-acetylglucosaminyltransferase activity of B3GnT7 is thought to be specific for GlcNAc-6-sulphated LacNAc-containing glycans [59,64]. It cannot be ruled out that the LEL signal for B3GnT7 may be due to some degree of cross-reactivity.

Poly-N-acetyl-lactosamine (polyLacNAc), or polylactosamine, is a disaccharide repeat [-3Galβ1-4GlcNAcβ1-]_n_ widely present in both N- and O- linked glycans on proteins and plays important roles in a variety of processes [61]. It is also found in glycosphingolipids of the lactoneo-series, and there are indications that long-chain glycosphingolipids are also involved in modulation of cellular interactions and immune cell activation [65]. In complex *N*-glycans, the core mannose may harbour up to five terminal GlcNAc residues that can be further extended into linear chains of polylactosamine. The range of *N*-glycan modifications of LRP6 induced by B3GnT activity may therefore be highly diverse, especially considering that multiple *N*-glycans on LRP6 are modified. Although we have shown that B3GnT2 modification promotes cell surface trafficking of LRP6 and that this likely represents one of the main mechanisms of how it regulates Wnt/β-catenin signalling, it remains to be seen whether ligand–receptor interactions and/or aggregation of LRP6 into Wnt signalosomes are affected. This should be investigated in future work.

Our loss-of-function data show that B3GnT2 is required for normal levels of LRP6/Wnt signalling and the reduced LRP6 signalling seen in cells lacking B3GnT2 is only restored to normal levels upon reintroduction of B3GnT2. Nevertheless, significant LRP6 signalling remains in cells lacking B3GnT2. This likely reflects a non-essential role of B3GnT2 for LRP6 activity and points more to its role as a regulatory factor. However, we show that different members of the B3GnT family appear to modify LRP6 and we cannot rule out functional redundancy by other members that may compensate for the absence of B3GnT2. It remains unclear why overexpression of LRP6 in the B3GnT2−/− cell line results in a slightly increased amount of LRP6 compared to wild-type cells; however, it displays reduced signalling. The distribution of LRP6 in different cellular compartments plays a crucial role in its activity: for example, it has been reported that lipid-raft-located LRP6 forms a complex with Wnt and Frizzled proteins and undergoes caveolin-mediated endocytosis to activate Wnt/β-catenin signalling, while LRP6 in nonlipid rafts is internalised with clathrin to inhibit Wnt/β-catenin signalling [66,67]. A recent study has also demonstrated a role for glycosphingolipids in the localisation and internalisation of TGFβR1 receptors within lipid raft domains [68]. In addition to the effect B3GnT2 may have on LRP6 interactions, it will therefore also be important to investigate whether B3GnT2 can regulate the redistribution of LRP6 in different cellular compartments.

We have identified two LRP6 glycosylation sites, N81 and N692, that are essential for the maturation of LRP6 during its biogenesis. *N*-glycans present on these sites may promote maturation of LRP6 in the ER/Golgi through their ability to aid interactions with glycosyltransferases or LRP6 chaperones such as MESD. LRP6 is known to require *N*-glycosylation for trafficking and membrane localisation [25,26], and MESD, a specific chaperone for LRP6, promotes LRP6 expression on the cell surface and enhances its signalling activity [69]. It has also been reported that Mest/Peg1 represses LRP6 maturation by inhibiting its glycosylation [25]. Interestingly, site N859 appears to inhibit LRP6 activity, since its removal enhances Wnt/β-catenin signalling. Our analysis also demonstrated that removal of the *N*-glycan at N859 results in increased levels of polylactosamine, suggesting that lack of an *N*-glycan here may promote the extension of polylactosamine chains on other *N*-glycans. This means that *N*-glycans present at N859 may inhibit the function and/or processing of nearby *N*-glycans on LRP6. The fact that the LRP6 N859Q mutant shows both enhancement of basal LRP6 activity (Figure 4D) and a slight reduction of B3GnT2-mediated activation of LRP6 activity (Figure 4E) suggests a complex role for this *N*-glycan in the regulation of LRP6 activity and further studies will be required to elucidate its precise function. Matoba et al. have reported that *N*-glycans located near the hinge area (between PE2 and PE3) of LRP6 alter the bending angle of the ectodomain which consequently affects its signalling activity [11]. It will be important to investigate in more detail how the *N*-glycan sites cooperate to regulate the conformation and activity of LRP6 using different LRP6 *N*-glycosylation site mutant combinations. Although our MS analysis detected modification at only a few of the *N*-glycan attached to LRP6, such as N81 and Asn433, it is highly likely, considering the difficulty in detecting such modifications, that other *N*-glycans on the ECD of LRP6 are also modified by B3GnT2. Our LEL lectin blot analysis as well as analysis of LRP6 glycosylation mutants support this.

Considering B3GnT2 has recently been reported to modify several ligands and receptors that alter T-cell activation via interaction with cancer cells [41], there may be a more general role for B3GnT2 in the regulation of cancer-relevant signalling events, such as Wnt/β-catenin signalling.

## 5. Conclusions

This study had identified glycosylation of LRP6, a co-receptor for Wnt, as a new regulatory mechanism for Wnt/β-catenin signalling. The B3GnT family of glycosyltransferases plays an important role in the extension of polylactosamine on the antenna of complex *N*-glycans attached to many membrane proteins and we have demonstrated that B3GnT2, in particular, modifies multiple LRP6 N-glycans to promote Wnt/β-catenin signalling. This is a significant finding that will have an impact on our understanding of cellular signalling in general, but also for research on disease states, stem cell maintenance, and cellular aging, all of which are known to be controlled to a large part through Wnt signalling.

## Figures and Tables

**Figure 1 cells-12-00863-f001:**
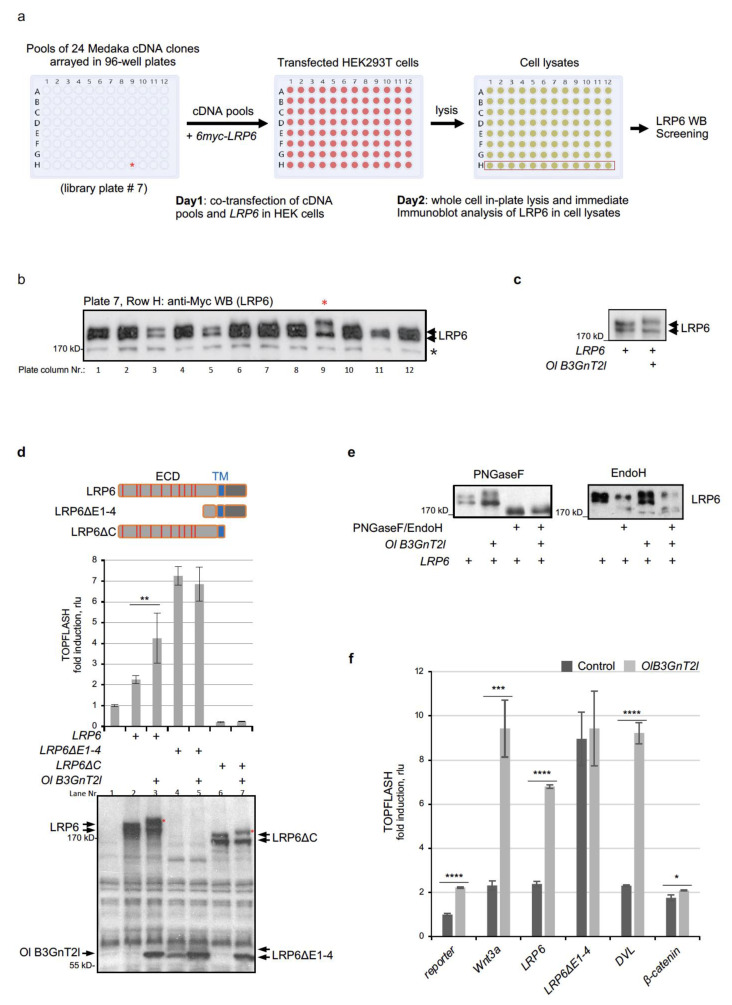
Identification of medaka B3GnT2-like glycosyltransferase as a novel LRP6 modifier. (**a**) Schematic overview of cell culture-based expression screening using pools of 24 medaka (*Oryzias latipes, Ol*) cDNA library plasmid DNAs, arrayed in a total of 8 x 96-well plates (cDNA pool plate #7 is shown). The cDNA pool in plate 7 at position H9, harboring *Ol B3GnT2*, is indicated by a red asterisk. (**b**) Original primary screen of LRP6 Western blot using the 12 cell lysates samples from row H of plate 7 (boxed row H of 96-well cell lysates plate shown in panel (**a**), showing the cDNA pool hit in lane 9 (red asterisk) that confers a clear upshift of the upper LRP6 band). A non-specific background protein band detected by the anti-Myc antibody is marked with a black asterisk. (**c**) Western blot screening of individual cDNA clones identified the single modifier as *Ol B3GnT2l* (XM_004075262). (**d**) TOPFLASH Wnt reporter assay (upper graph) and SDS_PAGE/WB (lower blot) of lysates from HEK293T cells transfected with the indicated plasmids in 96-well format. Amounts transfected: *LRP6*/*LRP6ΔE1-4*/*LRP6ΔC*, 20 ng; *Ol Flag-B3GnT2l*, 5 ng. Schematic depiction of the different LRP6 proteins expressed are shown to the right of the TOPFLASH graph. Red arrows on WB indicate the two protein bands corresponding to LRP6ΔE1-4, the lower of which overlaps with the Flag-B3GnT2l protein band. Asterisks indicate upshift of the upper protein bands for LRP6 and LRP6ΔC, whereas no upshift was detected for the LRP6ΔE1-4 upper band. (**e**) Western blots of cell lysates from HEK293T cells transfected with indicated plasmids in 96-well format. PNGaseF or Endo H were added to lysates in Laemmli loading buffer and incubated for 2 h at 37 °C. Amounts transfected: *LRP6*, 20 ng; *Ol Flag-B3GnT2l*, 5 ng. (**f**) TOPFLASH Wnt reporter assay showing effect of Ol B3GnT2l on Wnt/β-catenin signalling activity upon pathway activation using the indicated components. Amounts transfected per 96-well: *Flag-B3GnT2l,* 15 ng; *Wnt3a*, 5 ng; *LRP6*, 20 ng; *LRP6ΔE1-4*, 20 ng; *DVL1*, 20 ng; *β-catenin*, 5 ng. LRP6ΔE1-4 was used as LRP6 specificity control as it lacks all *N*-glycan sites. Data represent mean ± SD. * *p* < 0.1; ** *p* < 0.01; *** *p* < 0.001; **** *p* < 0.0001.

**Figure 2 cells-12-00863-f002:**
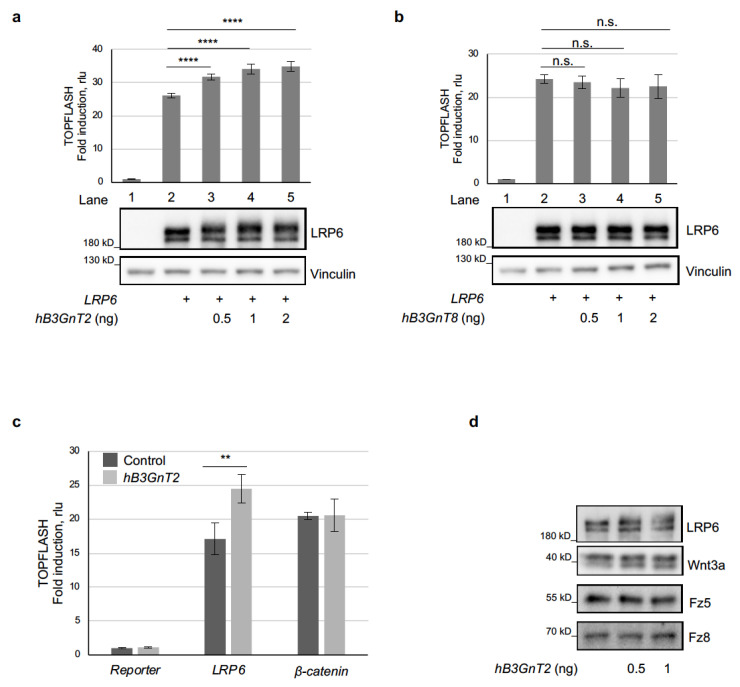
Human B3GnT2 promotes Wnt/β-catenin signalling and acts at the level of LRP6. (**a**,**b**) TOPFLASH Wnt reporter assays (upper graphs) and Western blots (lower blots) showing that human B3GnT2, but not B3GnT8, modifies LRP6 and promotes Wnt/β-catenin signalling in a dose-dependent manner. Amounts transfected: *LRP6*, 20 ng; *hB3GnT2* or *hB3GnT8*, as indicated. Data represent mean + SD. **** *p* < 0.0001. n.s., not significant. (**c**) TOPFLASH reporter assay in HEK293T cells in 96-well format. Amounts transfected: *LRP6*, 20 ng; *β-catenin*, 50 ng; *hB3GnT2*, 0.5 ng. Data represent mean + SD. ** *p* < 0.01 (**d**) Western blots of cell lysates from HEK293T cells in 96-well format. Amounts transfected: *hLRP6*, 20 ng; *mWnt3a*, 8 ng; *V5-mFz5* or *V5-mFz8*, 20 ng; *hB3GnT2*, as indicated.

**Figure 3 cells-12-00863-f003:**
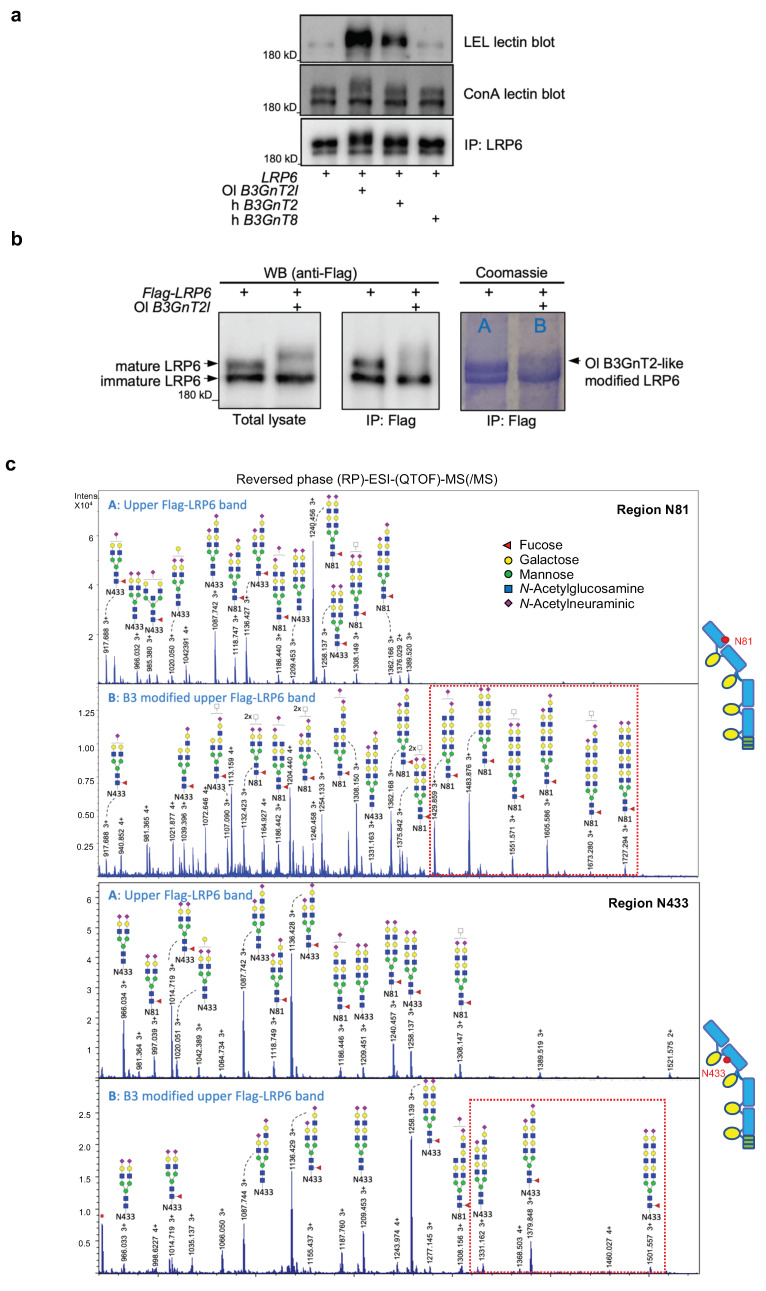
Ol B3GnT2l and human B3GnT2 modify LRP6 via polylactosamine synthesis. (**a**) Western blot and lectin blot analysis of immunoprecipitates (IP) from HEK293T cells transfected as indicated in 6-well plate format. Amounts transfected: *hLRP6*, 600 ng; *mMesd*, 150 ng; indicated *B3GnTs*, 30 ng. Lycopersicon esculentum lectin (LEL) was used to detect the repeating disaccharide units of polylactosamine. Concanavalin A (ConA) lectin was used to detect α-mannose cores within *N*-glycan chains. (**b**) Samples prepared for Mass Spectrometry (MS) analysis. Small aliquots were removed for total lysates or immunoprecipitates (IP) for Western blot analysis and the remaining IP samples were loaded on an SDS/PAGE gel followed by Coomassie blue staining. The upper bands from the Coomassie gel (A and B) were excised and subjected to Mass Spectrometry analysis. (**c**) Mass spectra in MS1 of charged glycans (disialylated) from LRP6 samples A and B analysed by (RP)-ESI-(QTOF)-MS(/MS). Sample A: upper Flag-LRP6 band. Sample B: Ol B3GnT2-like modified upper Flag-LRP6 band. Note modification of *N*-glycan chains (red dashed box in spectrum B) attached to Asn81 and Asn433 of LRP6, the position of which is indicated schematically on right.

**Figure 4 cells-12-00863-f004:**
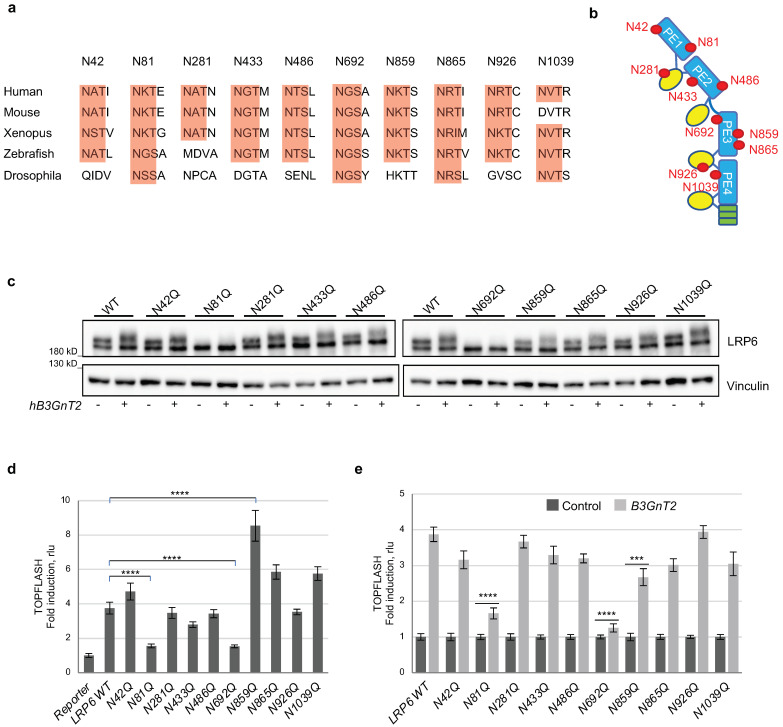
Multiple N-glycan sites of LRP6 are relevant for B3GnT2-mediated LRP6 promotion. (**a**) Evolutionarily conserved *N*-glycosylation sites. Red: asparagine (N) residues predicted to be *N*-glycosylated, with indicated amino acid position above. (**b**) Schematic model of LRP6 extracellular domain and its *N*-glycosylation sites. *N*-glycosylation sites are indicated as red filled circles: PE1 (N42, N81, N281); PE2 (N433, N486); PE3 (N692, N859, N865, N926); PE4 (N1039). PE, β-propeller/EGF-like structural domain. (**c**) Western blots of lysates from B3GnT2−/− HEK293T cells transfected with indicated *hLRP6* constructs (20ng) alone or with *hB3GnT2* (2 ng) in 96-well format. (**d**,**e**) TOPFLASH reporter assay using lysates of B3GnT2−/− HEK293T cells transfected with indicated *hLRP6* constructs alone or with *hB3GnT2* shown in (**c**) in 96-well format. (**d**) Wnt signalling activity of samples expressing LRP6 alone. Data represent mean ± SD. **** *p* < 0.0001. (**e**) Wnt signalling activities of samples expressing LRP6 alone shown in (**d**) are set as 1 and activities of samples expressing LRP6 and hB3GnT2 are normalised to the corresponding samples expressing only LRP6. Data represent mean ± SD. *** *p* < 0.001, **** *p* < 0.0001.

**Figure 5 cells-12-00863-f005:**
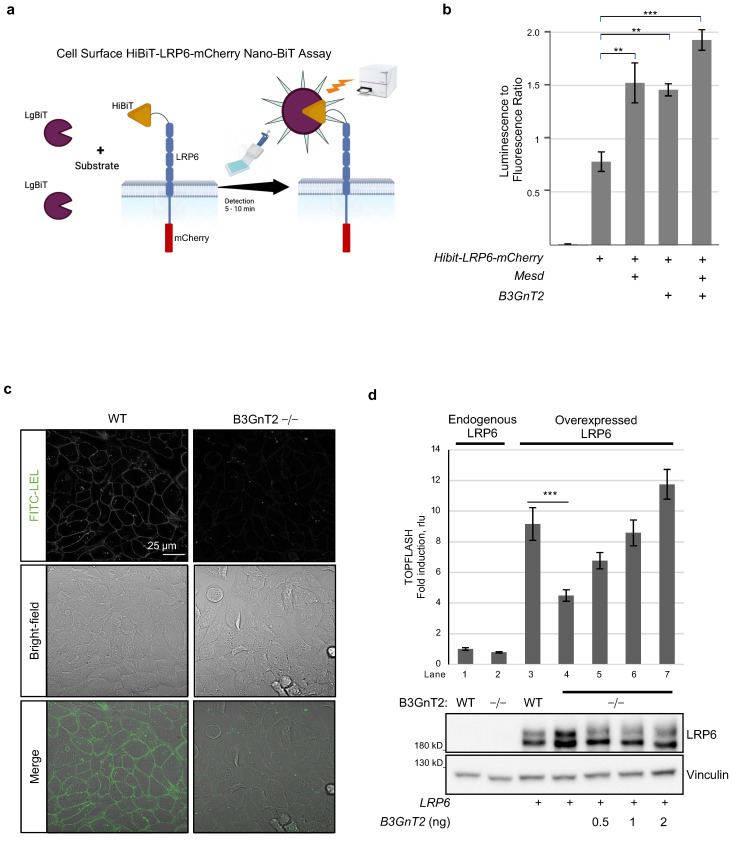
Human B3GnT2 promotes cell surface LRP6 and is required for Wnt/β-catenin signalling. (**a**) Schematic overview of cell surface LRP6 assay is shown on left. (**b**) Cell surface luminescence/fluorescence assay for HiBiT-LRP6-mCherry. HEK293T cells in 96-well plates were transfected as indicated with *HiBit-hLRP6-mCherry* (20 ng), *mMesd* (5 ng) and *hB3Gnt2* (0.5 ng). Cells were washed after 48 h with HBSS and treated with detection reagent constituting LgBiT (1:200), Fumarazine (1:100), HEPES (1:100) FBS (5%) and phenol free DMEM medium. The bioluminescence from reconstituted HiBit/LgBiT (Nluc) was measured using a Caloristar multiplate reader and the ratio of bioluminescence to fluorescence was plotted as shown in the graph. Data represent mean ± SD. ** *p* < 0.01; *** *p* < 0.001. (**c**) Confocal microscopy analysis of cell surface polylactosamine using FITC-LEL. Note that the cell surface polylactosamine level of B3GnT2−/− cells is much lower than HEK293T WT cells. (**d**) TOPFLASH reporter assay (upper graph) and Western blot (lower panels) of lysates from HEK293T WT or B3GnT2−/− cells transfected as indicated in 96-well format. Amounts transfected: *hLRP6*, 20 ng; *hB3GnT2*, as indicated. Note that the deficiency in signalling activity in B3GnT2−/− cells is rescued by re-expression of human B3GnT2. Data represent mean ± SD. *** *p* < 0.001.

## Data Availability

Not applicable.

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
