# Peer review of "N-Glycosylation of LRP6 by B3GnT2 Promotes Wnt/β-Catenin Signalling"

_cells, 2023, doi:10.3390/cells12060863_

Round 1
Reviewer 1 Report
This article by Xu et al. described a mechanism of LRP6 regulation by the glycosyltransferase B3GnT2. The authors showed that B3GnT2 leads to glycosylation of LRP6 at specific sites in the extracellular domain. This further induces the activation of Wnt signaling pathway. However, revisions are required to improve the study.
1. The introduction is a little lengthy. Authors only need to focus on the knowledge relevant to the study. Discussion should also be more focused.
2. Is there any evidence showing B3GnT2 directly regulate LRP6? For example, do they interact with each other? Is there any in vitro glycosylation experiment that can be done?
3. In Fig 1, authors showed that Ol B3GNT2l up-regulates the "glycosylation" of LRP6. I understand that authors transfected glycosyltransferases. However, authors should prove that the elevated band is solely due to glycosylation instead of other modifications such as phosphorylation.
a. In 1b, the levels of LRP6 vary between columns. The molecular weight seem to vary too, like in column 11. Do authors have a better blot to minimize these variables?
b. Author should include at least one negative control in 1c.
c. Wnt ligands bind to the ECD of LRP6. But when over expressing LRP6 delE1-4, authors showed a significant increase of Wnt reporter activity, double that of LRP6 WT. Can authors explain why? Also, the immunoblot is quite messy...It's difficult to see the delE LRP6 mutant, not to say distinguish it from Ol B3GnT2l. Authors need to show a clear blot.
d. In 1e, the treatment of PNGaseF lead to bands at ~170 kd, which is likely the unmodified form of LRP6 (also seen in 1b). However, authors mentioned in the introduction that there should be only two bands for LRP6. Which one is true?
e. In 1e, the levels of each band also showed a significant decrease in EndoH treated samples. This indicates EndoH also affects mature LRP6, which is inconsistent with what authors wrote.
4. In Sup fig 2 and 3, it seems like other hB3GnTs are also modulating LRP6 glycosylation, such as hB3GnT3 and 4. However, this is not seen in the screening. Does it indicate the functional difference between Ol and human B3GnT? Authors should take this into consideration, at least include something in introduction or/and discussion.
5. In Fig 2c, authors concluded that B3GnT2 acts upstream of bcatenin. However, bcat is downstream of LRP6, anything that happens upstream should not affect downstream effects anyways. This does not support the epistasis of B3GnT2 function, not to say that so many Wnt regulators are in between LRP6 and bcat, such as dvl and the destruction complex components. As mentioned in point #2, authors need evidence supporting a direct mechanism. Or inhibit something upstream of LRP6 to address the epistasis.
6. Fig 2d: I don't understand the point of this experiment. Do Wnt3a, Fz5 and 8 all go through glycosylation? Authors should try looking at some functional aspects, such as the assembly of the Wnt membrane signalosome or recruitment of dvl.
7. Fig 3 and 4, again, showed a difference between Ol and human B3GnT2, with the fact that N81 is the conserved site. Authors should at least mention something in the discussion.
8. Authors showed nice data to support the increase of cell surface LRP6 by B3GnT2.
a. Authors should do statistical analyses in 5b.
b. In 5c, can authors also include a cell surface marker?
Overall, I suggest to accept this work after a major revision.
Reviewer 2 Report
The manuscript evaluates the post-translational mechanisms of the LRP6 co-receptor extracellular domain, namely the role of the B3GnT2 N-glycosyltransferase as a positive regulator of LRP6 that regulates β-catenin dependent Wnt signaling. The author's main conclusions are supported by gain-of-function and loss-of-function experiments. This is an extremely well-conducted and high-quality study that contributes to the elucidation of the regulation of the LRP6 co-receptor in the Wnt/β-signaling. Below are some minor considerations related to aspects that are less clear in the manuscript.
Minor considerations:
1- The manuscript describes an extensive set of experiments and results, with also an extensive introduction and discussion. To facilitate the analysis of the manuscript, I suggest that the introduction and discussion be shortened (while still including information essential for the interpretation of the results).
2- Line 121: Give the reference of the mouse monoclonal anti-V5 antibody.
3- Line 280: Figure 1. The positive cDNA pool (where the mature form of LRP6 was detected) is identified as H9 but should be lane 9, as the image may give the impression that this pool is at position I9.
4- Figure 1d: Although RP6ΔE1-4 and OlB3GnT2l co-expression does not lead to an increase in Wnt/β- catenin activity when compared with RP6ΔE1-4 alone (lanes 4, 5), the levels of activation with RP6ΔE1-4 and Ol B3GnT2l co-expression are higher/similar to the activation for co-expression of LRP6 and OlB3GnT2l. How can this result be explained? The results obtained in lanes 6 and 7, in my opinion; provide more information on the role of OlB3GnT2l in activating the LRP6/Wnt-β-catenin signaling.
5- Figure 3a. The results of the Con A and LEL blots, which detect the N-glycosylated LRP6 and polyLacNAC extensions levels, as the authors point out, seem to demonstrate that the cells that co-expressed LRP6 and OlB3GnT2l promote a higher level of N-glycosylation and polyLacNAC extensions, compared with the co-expression of LRP6 and OlB3GnT2. The significance of this result has not been explored. What are the possible explanations for this result, since the results in figure 5c show that B3GnT2 is the main responsible for polylactosamine synthesis and, consequently, for LRP6 activity?
6- Figure 5d: Interestingly, the level of pathway activation (Figure 5d) in the B3GnT2 -/- cells with overexpression of LRP6 (Lane 4) is considerable (well above the basal level; Lane 1 and 2). In this case, there is a positive response (lane 2 vs 4), increased activation, in the presence of LRP6 (although at lower levels than WT with overexpression of LRP6). Could this result indicate some functional redundancy of B3GnTs in the absence of B3GnT2 (which should be preferential)? There are one or other of these issues that are covered briefly in the discussion but I think they can be explored/discussed more concretely (not just a brief description of the results), and especially in the results they should be briefly mentioned.
Round 2
Reviewer 1 Report
Authors have considered all the points carefully and included some additional experimental proof.
1. I suggest authors to substitute current Figure 2C with reviewer figure 4.
2. Reviewer figure 2 and 3 can serve as a supplement.
